Effects of mining on the molybdenum absorption and translocation of plants in the Luanchuan molybdenum mine

Yin Kejing 1
Shi Zhaoyong 9903105@haust.edu.cn 1 2 3 4
Zhang Mengge 1
Li Yajuan 1
1 College of Agriculture, Henan University of Science and Technology , Luoyang , Henan Province , China
2 Luoyang Key Laboratory of Symbiotic Microorganism and Green Development , Luoyang , Henan Province , China
3 Henan Engineering Research Center of Human Settlements , Luoyang , Henan Province , China
4 Key Laboratory of Mountain Surface Processes and Ecological Regulation , Chengdu , Sichuan Province , China
Gao Junkuo
Electronic publication date: 2020 May 26
Publication date: 2020
Volume: 8
Electronic Location ID: e9183
Received 2020 Feb 6; Accepted 2020 Apr 22
Copyright: ©2020 Yin et al.
Copyright year: 2020
Copyright holder: Yin et al.
License: This is an open access article distributed under the terms of the Creative Commons Attribution License, which permits unrestricted use, distribution, reproduction and adaptation in any medium and for any purpose provided that it is properly attributed. For attribution, the original author(s), title, publication source (PeerJ) and either DOI or URL of the article must be cited.
License URL: https://creativecommons.org/licenses/by/4.0/

Keywords: Molybdenum mining, Mo concentration, Absorption, Translocation, Soil pollution

Funding: Program for Science & Technology Innovation Talents in Universities of Henan Province 18HASTIT013 NSFC 31670499 Scientific and technological research projects in Henan province 192102110128 Key Laboratory of Mountain Surface Processes and Ecological Regulation 20160618 Laboratory for Earth Surface Processes, Ministry of Education 201612 The Innovation Team Foundation (2015TTD002) of Henan University of Science and Technology This research was funded by the Program for Science & Technology Innovation Talents in Universities of Henan Province (18HASTIT013), NSFC (31670499), the Scientific and Technological Research Projects in Henan province (192102110128), the Key Laboratory of Mountain Surface Processes and Ecological Regulation, CAS (20160618), the Laboratory for Earth Surface Processes, Ministry of Education (201612), and the Innovation Team Foundation (2015TTD002) of Henan University of Science and Technology. The funders had no role in study design, data collection and analysis, decision to publish, or preparation of the manuscript.

==============================
Background

There is a critical need to examine whether mining of molybdenum (Mo) ore will affect Mo absorption and translocation by plants at a community level.

Methods

Indigenous plants and their rhizospheric soil (0–20 cm) growing in two different areas including the mining and the unexploited areas were collected from the Luanchuan Mo mine—one of the largest Mo mines in Asia. The concentrations of Mo and other heavy metals of plants or soil were measured by ICP-AES. Mo absorption and translocation in plants growing in two areas were investigated and compared. Heavy metal pollution in soil was also evaluated by the potential ecological hazard index method.

Results

Mo concentration in mining soils was higher with the changes from 108.13 to 268.13 mg kg−1 compared to unexploited area. Mo concentrations in shoots and roots of plants growing in the mining area were also significant higher than those growing in the unexploited area with 2.59 and 2.99 times, respectively. The Mo translocation factor of plants growing in the unexploited area was 1.61, which reached 1.69 times that of plants growing in the mining area. Mo was the main heavy metal pollutant in the soil of both the mining and the unexploited areas.

Conclusion

Mining of Mo had changed not only the Mo concentration in soil but also Mo absorption and translocation in plants. Plants growing in the mining area absorbed more Mo from the soil but translocated relatively less to shoots than plants of the unexploited area. However, the mechanisms of Mo absorption and translocation of plants in mining area should be further studied in the future.

Introduction

Mo is an important mineral resource, used in metallurgy, chemical industry, aerospace industry, and medicine, among other fields (Li et al., 2009). By the end of 2014, China’s molybdenum reserves were 4.3 million tons, accounting for 39.09% of the world’s total. The annual consumption of molybdenum is 75,000 tons in China, accounting for more than one third of the world’s total consumption (Zhang et al., 2017). Mining production has increased economic growth in Mo mines, but varying degrees of pollution have been caused by long-term mining activity (Yu et al., 2011;Yu et al., 2012; Wang et al., 2018a; Wang et al., 2018b; Wang et al., 2018c; Zeng et al., 2019).

Yu et al. (2018) found that the molybdenum content in drinking water of the Luanchuan reservoir reached a level of serious pollution, and the concentration of Mo exceeded the standard limit by 2.44 times. Jia et al. (2015) showed that the Mo concentration is 87 times higher in mining area than unexploited area in Fujian province. Zhang et al. (2019) suggested that Mo mining increased Mo accumulation in adjacent croplands to lead to 3.57 times in rice than that of Mo threshold. In a conclusion, the mining of Mo resulted in accumulation in soils and plants (Qu et al., 2008; Huang et al., 2011).

The effects of Mo pollution in soil-plant systems has attracted much attention. Many researchers have confirmed that a large amount of exogenous Mo in soil may enhance Mo absorption by plants (Boojar & Tavakkoli, 2011; Wang et al., 2018a; Wang et al., 2018b; Wang et al., 2018c; Shi et al., 2018). Boojar & Tavakkoli (2011) found Mo concentrations in shoots and roots of Achilla tenuifolia were 1,769 mg kg−1 and 210 mg kg−1 when Mo concentration in soil reached 448 mg kg−1. Wang et al. (2018a), Wang et al. (2018b) and Wang et al. (2018c) found Mo concentration in shoots and roots of Macleaya cordata to be as high as 704.4 mg kg−1 and 398.9 mg kg−1, respectively, when soil Mo concentration was 256.1 mg kg−1. Shi et al. (2018) found thet Mo concentration reached 786.9 mg kg−1 and >900 mg kg−1 in maize shoots and roots, respectively, when 1,000 mg kg−1 Mo was added to soil. Mo is required in only small amounts by plants to maintain normal growth and development, but chlorosis and yellowing can be caused by excess Mo (Hale et al., 2001; Tow et al., 2016; Niu et al., 2019). At present, there is no clear definition of the toxicity threshold of molybdenum in plants, as different species have different Mo tolerances (Gupta, Chipman & Mackay, 1978; Huang et al., 2003; McGrath et al., 2010; Li, 2016). Previous studies found that the threshold of Mo poisoning was 192 mg kg−1 in Trifolium pretense (Gupta, Chipman & Mackay, 1978), 100–200 mg kg−1 in Lactuca sativa (Gupta, Chipman & Mackay, 1978) and Solanum tuberosu (McGrath et al., 2010), 500 mg kg−1 in Triticum aestivum (Li, 2016), 640 mg kg−1 in Brassica oleracea (Gupta, Chipman & Mackay, 1978), and more than 1,000 mg kg−1 in Lycopersicon esculentum (Huang et al., 2003).

Different translocation mechanisms could be used by plants to resist excess Mo (Kádár, Koncz & Gulyás, 2000; Lian, Xu & Han, 2011; Kovács et al., 2015). Kádár, Koncz & Gulyás (2000) found that maize can absorb appreciable quantities of Mo from the soil by roots, but Mo translocated from roots to shoots decreased with the increase of Mo concentration in soil. Similar results were also found by Kovács et al. (2015). Lian, Xu & Han (2011) found Phragmites australis stored Mo in roots and transferred little Mo to shoots, but Typha orientalis translocated large quantities of Mo from roots to shoots.

Although extensive Mo mining has created great material wealth in some areas, negative effects of Mo pollution on local soil-plant systems may be substantial (Haque et al., 2008; Boojar & Tavakkoli, 2011). There are few studies about Mo absorption and translocation of indigenous plants growing in Luanchuan. Further, many researchers have focused on Mo absorption and translocation of cultivated plants (Gupta, Chipman & Mackay, 1978; McGrath et al., 2010; Kovács et al., 2015), and whether Mo mining activity will increase Mo absorption of single or a few chosen species (Boojar & Tavakkoli, 2011; Wang et al., 2018a; Wang et al., 2018b; Wang et al., 2018c). It is unclear how molybdenum mining will affect Mo absorption and translocation for entire indigenous plant communities.

In this study, we sampled indigenous plants growing in a mining area (400 m around the center of the mining area) and an unexploited and proven mining area (2,000 m from the edge of an unexploited molybdenum mine) of the largest molybdenum mine of Asia. Mo absorption and translocation of plants growing in the mining and the unexploited areas were compared, which would provide a scientific basis for ecological restoration and vegetation reconstruction of this and similar mine areas. Therefore, our hypothesis is that mining of Mo will increase Mo concentration in soil, which leads to Mo accumulation in plants.

Materials & Methods

Study area

In Luanchuan County (33°46′N, 111°37′E), reserves of molybdenum mines are 2.06 million tons, ranking first in Asia and third in the world (Wang & Tian, 2000; Zeng et al., 2013). The development of molybdenum mining in Luanchuan began in 1960 (Wang & Tian, 2000). There are 27 open pit mines, 9 mines are still in production, and quantities of daily mining can reach 13,000 tons (Wang & Tian, 2000; Huo, Yang & Zhang, 2007; Zeng et al., 2013). To confirm whether mining will affect Mo absorption and translocation in plants or not, we selected two representative areas in the Luanchuan molybdenum mine area for this study. The mining area (33°53′–33°55′N, 111°27′–111°29′E) was located at the edge of the central mining area (around 400 m away from the center of the mining area), and it had been affected by mining, but the ground was still covered with vegetation. Mo reserves are 1.33 million tons in the central mining area with the mining quantities of 5000 tons per day (Wang & Tian, 2000). The unexploited area (33°48′–33°53′N, 111°27′–111°30′E) was located in a proved and not mining Mo mine, which are away from more than 2,000 m of the edge mining area.

Sample collection

Sample belts were established in the mining and the unexploited Mo mine areas.. In each sample belt, five 20 × 20 m plots were set with 50–100 m apart. At every sample plot, plants were identified and recorded into species by a botanist (Five mature individuals of each plant species were collected if the number of a plant species was more than 5; Table 1). Plant species were identified as dominant when the number of times of appearance of each species was greater or equal to 3 in all five plots of each study area. Of the plant species in the mining area, 22 plant species are collected, which belong to 18 genera of 10 families (Table 1). As far as the unexploited area is concerned, 23 plant species belonging to 20 genera and 9 families were recorded (Table 1). The nine shared species are observed in the mining and the unexploited areas including Ailanthus altissima, Artemisia argyi, Artemisia capillaris, Lespedeza bicolor, Phragmites australis, Poa annua, Robinia pseudoacacia, Setaria viridis, Sorghum propinquum, which reach 41% and 39% of total plant species in the mining and the unexploited areas. The dominant herbaceous plants in both the mining and the unexploited areas are the familis of Gramineae and Compositae, while the dominant family of woody plants is Leguminosae. When the plant communities are considered, herbaceous in the families of Gramineae and Compositae are dominant in two areas. Further, during our investigation, Lespedeza bicolor and Setaria viridis are dominant plant species in either mining or unexploited areas. Additionally, corresponding rhizosphere soil (0–20 cm) from each species was collected and mixed into a single soil sample for each plot.

Table 1 The shared species and endemic species in the mining and the unexploited area.

		Mining	Unexploited	
Shared species	Woody	Ailanthus altissima, Lespedeza bicolor, Robinia pseudoacacia	
	Herbaceous	Artemisia argyi, Artemisia capillaris, Phragmites australis, Poa annua, Setaria viridis, Sorghum propinquum	
Endemic species	Woody	Amorpha fruticosa, Cotinus coggygria, Euonymus alatus, Euonymus fortunei, Ligustrum lucidum, Rosmarinus officinalis, Salix linearifolia	Albizia julibrissin, Indigofera tinctoria, Juglans regia, Picrasma quassioides, Pinus massoniana, Populus tomentosa, Quercus acutissima, Toxicodendron vernicifluum	
Herbaceous	Asparagus cochinchinensis, Artemisia sacrorum, Artemisia scoparia, Ophiopogon japonicus, Panicum bisulcatum, Picris hieracioides	Artemisia annua, Artemisia selengensis, Bidens pilosa, Dendranthema indicum, Medicago falcata, Triarrhena sacchariflora	

Measurement and calculation of heavy metals in soil and plants

Soil samples were air–dried at room temperature and sieved through two mm filters to remove plant roots and other material before analysis. The plant samples were cut into shoots and roots, they were carefully washed with deionized water and oven–dried at 105 °C for 30 min and 60 °C for 72 h, then ground into a fine powder, and sieved through one mm nylon sieve. Concentrations of Mo, Cu, Mn, and Zn were determined by ICP–AES (Agilent 720, California, USA).

Total nitrogen (STN), total phosphorus (STP), and organic matter (SOM) of soil were measured according to the methods of Bao (2000). The STN in the mining area and the unexploited area were 0.34 and 1.40 g kg−1, respectively. The STP were 0.75 and 0.79 g kg−1, respectively. The SOM were 17.47 and 22.16 g kg−1, respectively.

The potential ecological risk index was used to assess heavy metal pollution in soils (Hakanson, 1980). The potential ecological risk index of a heavy metal is Ei = Ti ×Ci. Ti is the toxic coefficient of heavy metals: Cu = 5 (Xu et al., 2008), Mn = Zn = 1(Xu et al., 2008), and Mo = 15 (Lin, 2009); Ci is the ratio of metal concentration in study area to the corresponding soil background value in China (Wei et al., 1991). The total potential ecological risk index of multiple heavy metals is RI = ∑ Ei. Evaluation criteria from Hakanson (1980) were followed. To assess Mo translocation in plants, a translocation factor (TF) was used. TFs were calculated as follows: TF = Concentration of Mo in shoots/ Concentration of Mo in roots (Kisku, Barman & Bhargava, 2000).

Statistical analysis

Heavy metal concentrations in the mining and the unexploited areas were compared using t-tests, and a value of P ≤ 0.05 was considered significant. A t-test was also employed to compare the difference in Mo absorption and translocation of plants growing in the mining and the unexploited areas at P ≤ 0.05. Linear regression was used to assess the relationship of Mo concentration between roots and shoots of plants. All analyses were conducted in SPSS 21.0 (Chicago, USA).

Results

Heavy metal concentration and potential ecological risk assessment

Four heavy metals of Mo, Cu, Mn and Zn were considered due to their higher concentration than others. Concentrations of Mo, Cu, and Mn were significantly higher in the mining area with values of 268.13 mg kg−1, 136.45 mg kg−1, and 1,282.46 mg kg−1 than in the unexploited area, respectively (Table 2). The Zn concentration was 456.22 mg kg−1 in the mining area compared to 370.70 mg kg−1 in the unexploited area. These results indicated that Mo mining had increased 2.48, 2.14, 1.36, and 1.23 times of Mo, Cu, Mn, and Zn concentration in mining soil than unexploited soils, respectively. Concentrations of the four heavy metals in the mining and unexploited areas were much higher than soil background values of China (Wei et al., 1991). The highest was Mo, which was 134.07 times higher in the mining area and 54.07 times higher in the unexploited area. Zn, Cu, and Mn reached 6.15, 6.04, and 2.20 times higher in the mining area, and 5.00, 2.82, and 1.61 times in the unexploited area. Zn and Cu concentrations in the mining or unexploited areas were greater than second grade thresholds set by State Environmental Protection Administration of China (1995) (GB15618-1995).

The Ei value of Mo reached 2010.98 and 810.97 in the mining and the unexploited areas, respectively, which indicated soils had been extremely contaminated by Mo on the basis of the evaluation criteria established by Hakanson (1980). All Ei values of Cu, Mn, and Zn were less than 40 in both mining and unexploited area, indicating that soils were slightly contaminated by them. Further, Mo accounted for 98.12 % and 97.54% of the total potential ecological risk indexes of the mining and the unexploited area according to the Ei value, respectively (Table 3). Overall, Mo is the main pollutant in both the mining and the unexploited area in soils of Luanchuan molybdenum mine areas.

Mo concentration of plants

Mo concentrations of shoots and roots were higher with the averages of 124.02 and 131.30 mg kg−1 in the mining area compared to 47.80 and 43.92 mg kg−1 in the unexploited area, respectively (Fig. 1). Mo concentration in roots and shoots of herbaceous plants growing in the mining area increased significantly by 3.58 and 2.27 times, respectively, compared to that in the unexploited area (Fig. 2). When the dominant plant species were considered, Mo concentration in roots and shoots of Lespedeza bicolor and Setaria viridis were significantly higher in the mining area than them in the unexploited area, respectively (Fig. 3). Our results indicated enhancement of Mo in roots and shoots of plants had been caused by mining activity of Mo mine.

Table 2 Soil heavy metal concentrations (mg kg1) in mining and unexploited areas.

	Mo (mg kg−1)	Cu (mg kg−1)	Mn (mg kg−1)	Zn (mg kg−1)	
Mining	268.13a	136.45a	1282.46a	456.22a	
Unexploited	108.13b	63.80b	938.14b	370.70b	
Threshold	–	≤100a	–	≤300a	
Background value	2.00b	22.60b	583.00b	74.20b	
Notes.

Different letters after the values of each column represent significant difference at P < 0.05 level, according to T-test.

a second grade threshold of the Environmental Quality Standard for Soils in China (GB15618-1995).

b Soil background values of China (Wei et al., 1991).

Table 3 The potential soil ecological risk indexes of heavy metals in Luanchuan molybdenum mine area.

Area	Ei	RI	
	Mo	Cu	Mn	Zn		
Mining	2010.98(98.12%)	30.19(1.47%)	2.20(0.11%)	6.15(0.30%)	2049.45	
Unexploited	810.97(97.54%)	14.12(1.70%)	1.61(0.19%)	5.00(0.60%)	831.39	
Notes.

The values in brackets represent the risk index of a single heavy metal as a percentage of the total risk index.

Figure 1 Mo concentration of plants growing in the mining and the unexploited areas.

Values represent mean ± standard error (SE), the same as below. An asterisk (*) above bars indicates significant differences between the same organs in plants for different areas at P < 0.05, according to the t-test.

Figure 2 Mo concentration of different life forms of plants growing in the mining and the unexploited areas.

An asterisk (*) above bars indicates significant differences between the same organs in the same plant life forms for different areas at P < 0.05, according to the t-test.

Figure 3 Mo concentration of dominant species growing in the mining and the unexploited areas.

An asterisk (*) above bars indicates significant differences between the same organs in the same species for different areas at P < 0.05, according to the t-test.

Mo translocation in plants

As shown by a t-test (Fig. 4), the Mo translocation factor of plants growing in the mining area was significantly lower with 0.95 compared to 1.61 in the unexploited area. Mo translocation in herbaceous plants growing in the mining area reduced significantly by 48.37%, compared to that in the unexploited area (Fig. 5). Similar trends were also observed in Mo translocation of Lespedeza bicolor and Setaria viridis growing in the mining area compared to the unexploited area, 0.95 compared to 2.18 and 1.02 compared to 2.07, respectively (Fig. 6). These results showed that Mo translocation of plants decreased with increasing Mo concentration in soil.

Figure 4 Mo translocation of plants growing in the mining and the unexploited areas.

An asterisk (*) above bars indicates significant differences between the plants for different areas at P < 0.05, according to the t-test.

Relationship of Mo between plant shoots and roots

The Mo concentration of shoot increased linearly with the enhancement of Mo concentration in roots (Fig. 7), which indicated that the distribution Mo in shoots and roots are determined by the root absorption. Further, the changing trends of Mo concentration in shoots with variation of roots are similar between the mining and the unexploited areas.

Discussion

Usually, mining activity of Mo could affect the soil Mo concentration (Haque et al., 2008). In this study, Mo concentrations in the mining area reached 268.13 mg kg−1 (Table 2), 2.48 times that in unexploited area, and 134.1 times the soil background value in China (2 mg kg−1, (Wei et al., 1991). This result showed that long-term mining activity had led to the enrichment of Mo in soil, which agrees with the previous findings. Han et al. (2019) confirmed that the Mo concentration reached 14.4 times the soil background value in Shaanxi Province because of the mining activity. Boojar & Tavakkoli (2011) demonstrated that Mo mining has resulted in a massive increase Mo concentration of soil and the mean concentration of Mo in the mining soil reached 448 mg kg−1. Similar results have been reported in other studies (Battogtokh, Lee & Woo, 2014; Tapia et al., 2018; Pipoyan et al., 2019). Further, Wang & Tian (2000) has confirmed that the associated heavy metal minerals with Mo in the Luanchuan molybdenum ore mainly include Cu, Mn and Zn. It is inevitable the long-term mining activity also caused the increase of Cu, Mn, and Zn in soil. Similar results have also been reported in the research of Han et al. (2019), who found Mo mining could not only increase the Mo concentration, but also increase the concentrations of the associated heavy metals such as Cu, Pb, and Zn in Shaanxi Mo mine area. Mo concentration in the unexploited area was 108.13 mg kg−1, 54.1 times higher than the background values (Table 2). The possible reason for the higher Mo in the unexploited area than soil background value is that Mo ore bodies in Luanchuan are shallow and exposed on the surface, which had been testified and showed suitable for large-scale, open pit mining (Wang & Tian, 2000; Zeng et al., 2013). The potential ecological risk assessment showed that the RI value in the mining area was as high as 2049.45, with the Mo Ei values of the mining area accounting for 98.12% of total potential ecological risk. These results suggest that Mo is the main pollutant in Luanchuan molybdenum mines.

Figure 5 Mo translocation of different life forms of plants growing in the mining and the unexploited areas.

An asterisk (*) above bars indicates significant differences between the same organs in the same plant life forms for different areas at P < 0.05, according to the t-test.

Figure 6 Mo translocation of dominant species growing in the mining and the unexploited areas.

An asterisk (*) above bars indicates significant differences between the same organs in the same species for different areas at P < 0.05, according to the t-test.

Figure 7 The relationship between shoot Mo and root Mo in plants.

(A) Mining area; (B) unexploited area.

The investigation of plant communities in the Luanchuan Mo mine area showed that plant communities are dominated by herbaceous Gramineae and Compositae both the mining and the unexploited areas (Table 1). This result indicated that herbaceous plants are more adaptable to the harsh environment of the mine area than woody plants, which consistent with the findings of Xue et al. (2014) who found that herbaceous plants have a higher resistance to environmental changes and they may play a dominant role in the plant community of mine areas. Meantime, we also found woody plants in both areas are dominated by Leguminosae, as these plants are likely to form a symbiotic relationship with rhizobia than woody of other families. This improves the nutrient acquisition capacity of Leguminosae and enhances their ability to persist in harsh environments (Zahran, 1999; Reed, 2017).

Generally, plants have a strong Mo tolerance, and most have no adverse reactions even when Mo concentration is greater than 100 mg kg−1 (Gupta, Chipman & Mackay, 1978; Huang et al., 2003; McGrath et al., 2010; Li, 2016; Shi et al., 2018). Boojar & Tavakkoli (2011) found that plants such as Achilla tenuifolia and Erodium ciconium could grow normally at the Sanj molybdenum mine, where the concentration of Mo in soil was 448 mg kg−1. In our previous maize culture pot experiment, we found maize could grow normally when 1,000 mg kg−1 Mo was added to soil (Shi et al., 2018). In the present study, Mo concentration in the Luanchuan Mo mine area exceeded normal ranges, which would generally be considered poisonous to plants, but no Mo toxic symptoms were observed for indigenous plants. This suggested that indigenous plants growing in the Luanchuan molybdenum mine area may have a strong adaption to Mo.

Mining not only leads to the enrichment of Mo in soil but also caused an increase of Mo concentration in plant tissues (Haque et al., 2008). In this study, Mo concentrations in the roots and shoots of plants growing in the mining area were 131.30 mg kg−1 and 124.02 mg kg−1, respectively, significantly higher than that in the same tissues of plants growing in the unexploited area (Fig. 1). Mo concentrations of Lespedeza bicolor and Setaria viridis growing in the mining area were significantly higher than that in the same plant species growing in the unexploited area (Fig. 3). A linear correlation indicated Mo in shoots increased with the concentration in roots (Fig. 7), which was similar to a previous study that indicated Mo addition significantly increased Mo concentrations in both shoots and roots of plants (Shi et al., 2018).

Overall, these results indicated mining had led to an increase of Mo absorption in plants, which is consistent with previous researches. Wang et al. (2018a), Wang et al. (2018b) and Wang et al. (2018c) observed mining resulted in the Mo concentration to 772.4 mg kg−1 in the shoots of Morus australis, which was 154 times than the contents in the plants growing unexploited area. Boojar & Tavakkoli (2011) revealed that Mo concentrations were greater than normal ranges in 36 plant species growing in a mining area, among which Achilla tenuifollia was the highest with 1,979 mg kg−1. Kumar & Singh (1980) found there is a synergistic relationship between Mo and sulfur (S) when the Mo supply level in soil is high. The main chemical composition of the Luanchuan molybdenum ore is MoS2 and the mining of Mo ore may cause exogenous S to enter the soil. Due to the high Mo concentration in the mining soil, S may play a major role in promoting the Mo absorption of plants. This is one possible reason that the Mo concentrations of plants growing in the mining area were significantly higher than that of plants growing in the unexploited area. In addition, no certain relationship on Mo concentration between soil and plant was observed (Fig. S1), which supported the previous findings (Li, 2016; Shi et al., 2018). Shi et al. (2018) showed Mo concentration in the shoots of maize was significantly lower in 2,000 mg kg−1 Mo treatment than treatments with 1,000 and 4,000 mg kg−1 Mo when plants were inoculated with arbuscular mycorrhizal fungi (AMF). But Mo concentration in the shoots and roots of maize did not change significantly with the increase of soil Mo concentration under or without AMF inoculation. Li (2016) revealed that Mo concentration in the shoots of two winter wheat varieties (Mo efficient and inefficient varieties) did not increase significantly as soil Mo concentration increased from 0.15 to 1,000 mg kg−1, while Mo concentration in roots increased significantly when soil Mo concentration increased from 500 to 1,000 mg kg−1. He also found Mo concentration of the grain in Mo efficient variety increased significantly as soil Mo concentration increased from 500 to 1,000 mg kg−1, but no significant increase in Mo inefficient variety. These results indicated that the relationship on Mo concentration between soil and plant may be affected by other factors such as soil microorganisms and plant species, which need to be studied further.

It has been testified that Mo translocation have huge range of changes in plants with environments and plant species (Kádár, Koncz & Gulyás, 2000; Haque et al., 2008; Lian, Xu & Han, 2011; Kovács et al., 2015; Qin et al., 2017; Ameh, Omatola & Akinde, 2019; Couto et al., 2019; Wang, Liu & Qin, 2019). Kovács et al. (2015) found the Mo translocation factor of maize decreased from 0.85 to 0.48 when adding 270 mg kg−1 of Mo to the soil. Lian, Xu & Han (2011) found the Mo translocation factors of Typha orientalis growing in nutrient solution with Mo concentrations 0–20 mg L−1 were always higher than 1, but the Mo translocation factors of Phragmites australis under the same Mo concentrations were always lower than 1. In our study, the Mo translocation factor of plants growing in the unexploited area was significantly higher than that in the mining area (Fig. 4), indicating the distribution of Mo in plant tissues changed with Mo concentrations in soil. Significantly lower Mo transolcation from roots to shoots was also found in herbaceous plants growing in the mining area (Fig. 5). Mo translocation in woody plants and dominant plant species both showed declining trends with increasing Mo concentrations in soil, although no significant differences were observed (Figs. 5 and 6). These results indicated plants growing in different areas likely have different translocation strategies to resist excess Mo. To reduce toxicity of molybdenum to plants’ physiological systems, plants growing in the mining area absorbed large amounts of Mo and stored it in their roots. In the unexploited area, Mo concentration in soil was relatively low, and plants growing in this area could absorb a large amount of Mo from soil and translocate it from roots to shoots.

The significant differences of Mo absorption and translocation in Figs. 2 and 4 between herbaceous and woody plants in the mining and unexploited areas may be explained by plant life form because previous studies found that different life form of plants have unequal absorption and translocation to excess Mo (Gupta, Chipman & Mackay, 1978; Huang et al., 2003; McGrath et al., 2010; Li, 2016). As to herbaceous plants, the marked variation in plant absorption and translocation of Mo between the mining and the unexploited areas may be determined by the responses of herbaceous plant to increase of Mo in soils. Shi et al. (2018) confirmed that Mo absorption and translocation of maize were significantly different under different Mo concentrations. Mo concentration in shoot and root of maize increased significantly 91.79 and 324.60 mg kg−1, respectively, but the Mo translocation in maize reduced significantly by 51.55%, when the soil Mo concentration increased from 2,000 to 4,000 mg kg−1. Similar results were also observed in winter wheat by Li (2016). However, the translocation of Mo in woody species is not significant difference between the mining and the unexploited areas, which is possibly correlated with plant characteristics and different elements. Zeng (2017) showed that the Cu translocations of the woody Albizia kalkora and Sophora japonica decreased with increase of Cu in soils based on four Cu levels, but Albizia julibrissin presented the opposite trend. Wang et al. (2018a), Wang et al. (2018b) and Wang et al. (2018c) also studied the translocation of four woody plants to Pb and Zn, which indicated that the translocations factors changed with woody plant species, element concentration and different elements. Therefore, we speculate that woody plants transfer Mo may be varied with plant species. Certainly, this needs to be further explored.

Many experiments have found that most plants not only have strong Mo tolerance but also have strong Mo accumulation capacity (Huang et al., 2003; Tow et al., 2016; Shi et al., 2018; Wang et al., 2018a; Wang et al., 2018b; Wang et al., 2018c). Huang et al. (2003) found Lycopersicon esculentum only presented chlorosis and yellowing when the Mo concentration in the shoots was as high as 1,000 mg kg−1. Tow et al. (2016) revealed the highest Mo concentration in yellow and green leaves of Axonopus compressus achieved 6,050 and 1,393 mg kg−1, respectively. In the present study, Mo concentration in shoots of Setaria viridis growing in the mining area was 164.69 mg kg−1, far higher than the normal Mo concentration in plants, indicating Setaria viridis has a strong ability to absorb Mo, which is similar to the findings of McGrath et al. (2010) and Shi et al. (2018). Slopes of the correlation of Mo between roots and shoots were not the same as for the Mo translocation factor of plants growing in the two areas. This may be caused by different plant communities or to different mechanisms of plant responses to Mo when Mo concentrations are enhanced in soils. Specific factors that cause such differences need to be further studied.

Conclusions

Mining of Mo has resulted in the importation of large quantities of Mo, Cu, Mn and Zn into soils. The concentrations of Mo, Cu, Mn and Zn in mining soil were 2.48, 2.14, 1.36 and 1.23 times of that in unexploited soil, respectively. The pollution assessment indicated that Mo was the main pollutant in both mining and unexploited soils of the Luanchuan mine area.

Mining of Mo increased the absorption and decreased the translocation of Mo in indigenous plants. The Mo concentrations in the shoots and roots in mining area are 2.59 and 2.99 times than them in unexploited area. The decrease of translocation in plant to Mo is mainly led by herbaceous species. The changes of Mo concentrations in soil are easier to cause the feedbacks of Mo translocation in herbaceous than woody plants.

This is the first study to investigate the effect of Mo mining of Mo absorption and translocation based on the indigenous plants in our knowledge. Our study will benefit for selecting suitable plants for phytoremediation of Mo contaminated soils. Additionally, further studies are needed to find out the specific relationship between soil Mo and plant Mo.

Supplemental Information

Figure S1 The relationship on Mo concentration between soil and plant

(a) Root; (b) Shoot.

Click here for additional data file.

Supplemental Information 2 Raw data applied for data analyses and preparation for Fig. S1 to present the relationship of Mo concentration between plants and soil in the mining area or the unexploited area

Click here for additional data file.

Data S1 Raw data for data analyses and preparation for Figs. 1–7 and Tables 1–2

Click here for additional data file.

Additional Information and Declarations

Competing Interests

Author Contributions

Data Availability

The authors declare there are no competing interests.

Kejing Yin and Zhaoyong Shi conceived and designed the experiments, performed the experiments, analyzed the data, prepared figures and/or tables, authored or reviewed drafts of the paper, and approved the final draft.

Mengge Zhang conceived and designed the experiments, analyzed the data, prepared figures and/or tables, and approved the final draft.

Yajuan Li conceived and designed the experiments, prepared figures and/or tables, and approved the final draft.

The following information was supplied regarding data availability:

The raw measurements are available in the Supplemental Files.

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
