# Peer review of "Effects of mining on the molybdenum absorption and translocation of plants in the Luanchuan molybdenum mine"

_PeerJ, doi:10.7717/peerj.9183_

## Round 0.1 · original submission · Major Revisions

Please revise according to the reviewers' comments.

Reviewer 1 ·

Basic reporting

1, The author make too simple and unclear conclusion at the end. This report warranted further study, but some specific guidance or suggestions for future strategy to solve Mo pollution problem should be provided.

5, The author should provide more clear definitions about mining area and unexploited area. For example, what is the scale of Mo production from the mining area? Does the molybdenum (Mo) ore both exposed to plants in both mining and unexploited areas?

Experimental design

2, Mo concentration in soil might not be linearly correlated with Mo accumulated in plants. In that case, author should further refer or study Mo concentration in soil and make comparison with Mo in plants.

Validity of the findings

3, “These previous studies suggest that mining gives rise to Mo accumulation 68 in soils (Qu et al., 2008;Huang et al., 2011)” should be ‘These previous studies suggested(demonstrated) that mining resulted an Mo accumulation of 68 in soils (Qu et al., 2008;Huang et al., 2011)”

4, “In this paper, we sampled indigenous plants growing near the largest molybdenum mine of Asia.” The term near is not clear, the author should provide exact distance.

Additional comments

The manuscript titled “Mining varied molybdenum absorption and translocation of plants growing in molybdenum mine area” by Yin et. al reported plants grown in mining area and unexploited areas and their Mo absorption and translocation at a community level. This study was meaningful for providing guidelines from relation of Mo concentration and their impacts on plants grown in mining area. The research was conducted in the Luanchuan Mo mine—one of the largest Mo mines in Asia and possible environmental factors were evaluated by the potential ecological hazard index. However, the author did not make strong and logical relations and some statements seems ambiguous to readers. Therefore, I suggest consideration for publication after the author revised the following parts:

1, The author should provide more clear definitions about mining area and unexploited area. For example, what is the scale of Mo production from the mining area? Does the molybdenum (Mo) ore both exposed to plants in both mining and unexploited areas?

2, Mo concentration in soil might not be linearly correlated with Mo accumulated in plants. In that case, author should further refer or study Mo concentration in soil and make comparison with Mo in plants.

3, “These previous studies suggest that mining gives rise to Mo accumulation 68 in soils (Qu et al., 2008;Huang et al., 2011)” should be ‘These previous studies suggested(demonstrated) that mining resulted an Mo accumulation of 68 in soils (Qu et al., 2008;Huang et al., 2011)”

4, “In this paper, we sampled indigenous plants growing near the largest molybdenum mine of Asia.” The term near is not clear, the author should provide exact distance.

5, The author make too simple and unclear conclusion at the end. This report warranted further study, but some specific guidance or suggestions for future strategy to solve Mo pollution problem should be provided.

Reviewer 2 ·

Basic reporting

no comment

Experimental design

no comment

Validity of the findings

no comment

Additional comments

This manuscript by Yin et. al investigated whether mining of molybdenum (Mo) ore would affect Mo absorption and translocation by plants at a community level. This research was attractive and important, considering from environmental contamination and remediation. Undoubtedly, the researchers performed a complex and laborious work, however, there are some major issues that need to pay attention and reconfirmation.


1. What about the soil conditions other than Mo concentrations for these two areas? Are they similar? From the list of plant species on these two areas, they are different. How are you sure they are comparable in the research?

2. Mo is the main heavy metal pollutant in soil for both the mining and unexploited areas. However, Mo could also be formed in different salts with containing of other elements such as (S)Sulfur. What specific types of Mo adsorbed? Are there any other elements detected?

3. It is ambiguous for conclusion that heavy metal (Mo) affected plants grown in mining areas with different mechanisms for Mo absorption and translocation. Since Mo is necessary for plant growth but only trace amount is needed, what is the tolerance range for Mo accumulation in plants?

4. Community level is needed further clarification.

5. Plants were mainly grown in 0–20cm soil, but is there representative depth (area) for the majority of Mo distribution?

6. Rational analysis with comparison from data from other places (like other Mo mining areas) will be helpful and solidify the conclusion, because the tile does not restrict the mine area.

7. The figure 3 is not cited in the paragraph. Please add it.

8. Please discuss the reason why the translocation factor of wood growing in mining area and unexploited area is barely different for woody, compared with Herbaceous.

---

## Round 0.2 · accepted · Accept

Based on the reviewers' comments. The manuscript is acceptable now.

Reviewer 1 ·

Basic reporting

no comment

Experimental design

no comment

Validity of the findings

no comment

Additional comments

This work is interesting and the authors have made this paper more readable and logic. Therefore, I suggest acceptance of this paper after all the necessary revisions.

Reviewer 2 ·

Basic reporting

no comment

Experimental design

no comment

Validity of the findings

no comment